# The Role of NLRP3, a Star of Excellence in Myeloproliferative Neoplasms

**DOI:** 10.3390/ijms24054860

**Published:** 2023-03-02

**Authors:** Elisa Parciante, Cosimo Cumbo, Luisa Anelli, Antonella Zagaria, Immacolata Redavid, Angela Minervini, Maria Rosa Conserva, Giuseppina Tota, Nicoletta Coccaro, Francesco Tarantini, Crescenzio Francesco Minervini, Maria Giovanna Macchia, Giorgina Specchia, Pellegrino Musto, Francesco Albano

**Affiliations:** 1Hematology Section, Department of Precision and Regenerative Medicine and Ionian Area (DiMePRe-J), University of Bari ‘Aldo Moro’, 70124 Bari, Italy; 2School of Medicine, University of Bari ‘Aldo Moro’, 70124 Bari, Italy

**Keywords:** myeloproliferative neoplasms, nucleotide-binding domain (NOD)-like receptor protein 3, inflammasome

## Abstract

Nucleotide-binding domain (NOD)-like receptor protein 3 (NLRP3) is the most widely investigated inflammasome member whose overactivation can be a driver of several carcinomas. It is activated in response to different signals and plays an important role in metabolic disorders and inflammatory and autoimmune diseases. NLRP3 belongs to the pattern recognition receptors (PRRs) family, expressed in numerous immune cells, and it plays its primary function in myeloid cells. NLRP3 has a crucial role in myeloproliferative neoplasms (MPNs), considered to be the diseases best studied in the inflammasome context. The investigation of the NLRP3 inflammasome complex is a new horizon to explore, and inhibiting IL-1β or NLRP3 could be a helpful cancer-related therapeutic strategy to improve the existing protocols.

## 1. Introduction

Myeloproliferative neoplasms (MPNs) are a set of uncommon, neoplastic blood disorders that affect the bone marrow. They are characterized by a set of mutations in the hematopoietic stem cells and progenitor cells (HSPCs), from which erythrocytes, leucocytes and platelets are derived. They are genetically extremely varied and exhibit an aberrant HSPC unregulated proliferation and/or an inhibition of the differentiation in the bone marrow (BM) [1,2,3]. Inflammatory conditions in the BM can lead to DNA mutations and genomic instability, which contribute to or cause the initial impact on the hematopoietic stem cells and trigger the clonal evolution-related mutations in the MPN [4]. The cancerous cell can change the niche to its advantage and at the expense of healthy HSPCs [5]. As a result, the blood cells proliferate out of control [6]. Polycythemia vera (PV), essential thrombocytemia (ET), primary myelofibrosis (PMF) and chronic myeloid leukemia (CML) are examples of MPNs. They are identified by a rise in erythrocytes, platelets, and bone marrow fibrosis, respectively [7]. CML, associated with the Philadelphia chromosome and sometimes even genomic deletions [8,9], is characterized by a clonal myeloproliferation, leading to a marked overproduction of both mature and immature granulocytes [10]. Some individuals develop PMF during the progression of PV and ET, which is frequently accompanied by complications such as thrombosis or hemorrhages. The worst possible outcome is the development of acute myeloid leukemia (AML) [6].

Somatic mutations in the *JAK2* gene (exon 12 or 14), the *CALR* gene (exon 9) or the *MPL* gene (exon 10) are found in MPNs. The point mutation in exon 14 of the *JAK2* gene is the most frequent genetic change that can be discovered in these entities (*JAK2V617F*) [11,12]. The MPL/JAK/STAT signaling pathway is constitutively activated by *JAK2V617F*, *MPL* mutations at position W515 (in the juxtamembrane domain), and pathologic *CALR* mutations that are out-of-frame insertions and/or deletions yielding a new C-terminal peptide [13,14]. In addition to these driver mutations, 10–15% of MPNs are classified as triple negative and typically have a worse prognosis since they lack any of these prevalent mutations [15]. Additional mutations contribute to the development of the disease by increasing the cell self-renewal and preventing differentiation, including those in the linker proteins (IDH2, SH2B3, and CBL), spliceosomal components (SRSF2, U2AF1, and SF3B1), epigenetic modifiers (specifically ASXL1, TET2, and EZH2), and metabolic modifiers [16]. In addition to the *JAK2*/*MPL*/*CALR* “driver” alterations, the latter are present in most MPNs [17]. Surprisingly, they seldom develop during the progression and frequently exist during the diagnosis [18]. This wide variety of genetic abnormalities in MPNs appears to contribute to the inflammasome activation. Depending on the mutational setting, the influence of the inflammasome on leukemogenesis can either promote or prevent leukemia. The context-dependent and tissue-specific character of such cytokines is probably the cause for their seemingly incongruous roles in tumor progression and antitumor immunity [19,20]. The presented work aims to investigate the role of NLRP3 in MPNs as a new horizon to ensure a more significant and accurate assessment of these disorders, paving the way for several novel therapeutic options.

## 2. Myeloproliferative Neoplasms: The Inflammatory and Immune Environment

In 2015, Hasselbalch and Bjørn combined the epidemiological, biochemical, pathogenetic, and clinical evidence, considering MPN as an inflammatory disorder—a paradigm of the relationship between chronic inflammation and oncogenesis [21]. Chronic inflammation is widely recognized as one of the main initiators of vascular damage, specifically endothelial injury [22]. The inflammatory state of the vascular system appears to be brought on by driver mutations, particularly those in the *JAK2* gene [23]. Chronic non-neoplastic inflammation caused by autoimmune diseases or recurrent infections leads to a steady release of pro-inflammatory cytokines such as TNFα, IL-6, and IL-8, as well as an accumulation of reactive oxygen species (ROS), which in turn promote the growth and spread of cancer by causing genetic instability and oxidative stress and blocking the apoptosis program and cell migration [24]. MPNs frequently result in a rise of these three inflammatory cytokines [23]. Among the pro-inflammatory ones, the most significant is IL-1β, which has been linked to the MPNs pathogenesis and its role in niche remodeling [25,26]. IL-1β modulates the gene expression related to the fever, vasodilation, and hypotension [27]. A recent study found that the ablation of IL-1β in MPN mice reduced the severity of the disease, concluding that IL-1β encouraged the clonal proliferation of HSPCs with the *JAK2* mutation [28].

Additionally, the *JAK2V617F* mutation, together with an inflammatory milieu, enhances MPN development. By secreting cytokines and activating the bystander immune cells, the malignant *JAK2V617F* mutant cells also contribute to the inflammatory microenvironment. the higher expression of the *JAK2V617F* mutation on the endothelial cells of MPN patients causes increased inflammation and permeabilization of the vascular bed, a reduction in the cell development and a more rapid cell senescence. This suggests that MPN cells change the immediate local environment to promote their growth and limit that of their normal counterparts [22,29,30,31]. Therefore, the inflammatory milieu induces death and cell cycle arrest in wild-type cells while favoring the proliferation of *JAK2V617F* mutant neoplastic hematopoietic stem cells. Hence, the relationship between chronic inflammation and MPN development becomes evident.

Recent studies have revealed that inflammation, based on the pathophysiology and development of myeloid malignancies, is mediated by the innate immune system. The innate immune system’s multiprotein cytosolic oligomers are inflammasomes and are responsible for triggering inflammatory reactions [27]. Myeloid cells, especially macrophages, are the primary cell types involved in the inflammasome assembly [32]. The prevalence of MPNs in patients was higher in those with autoimmune or inflammatory conditions [33,34]. The innate immune system interacts with various pattern recognition receptors (PRRs) to find the microbial contamination or tissue injury. PRRs can identify features that are common to multiple microbial species or endocrine substances produced by cell and tissue injury, both of which are referred to as pathogen-associated molecular patterns (PAMPs) and danger-associated molecular patterns (DAMPs) [35]. The PAMPs or DAMPs are recognized by monocytes, macrophages, neutrophils, and dendritic cells [36]. There are four distinct classes of PRR families, which include cytoplasmic proteins such as retinoic acid-inducible gene (RIG)-I-like receptors (RLRs), nucleotide-binding oligomerization domain (NOD)-like receptors (NLRs), transmembrane proteins such as the toll-like receptors (TLRs) and C-type lectin receptors (CLRs) [35]. The TLRs and their signaling pathways molecules TNFR1, TNFR2, and CD95, as well as the other important innate immune regulators, are upregulated or constitutively activated in HSPCs [37,38]. This suggests that these molecules play a significant role in developing myeloid malignancies. The dysregulation of these molecules causes aberrant hematopoiesis, imbalanced cell death, and a proliferation in patients’ bone marrow [39]. Among them, the NLRs appear to have a more prominent role in persistent non-infectious sterile inflammation than any other innate immune receptor molecule [40]. Most of these can assemble into the inflammasome complexes, including NLRP3, NLRP1, NLRP6, NLRC4, NAIP, AIM2, and pyrin [41], but NLRP3 is the undisputed protagonist of the inflammasome, involved in numerous cancers. Cell death and inflammation are controlled by the NLRP3 inflammasome regulation [42]. NLRP3 inflammasomes are also essential for tumor-specific adaptive immunity. As evidence of this, the lack of a functioning NLRP3 inflammasome in the mouse model failed the CD8+ T cell priming [43]. The analyzed data showed a close relationship between NLRP3 inflammasomes and carcinoma susceptibility, progression, and prognosis [44]. Inflammation, vascular damage, and dysimmunity were all strongly correlated. It has long been understood how the innate immune system and the equilibrium of pro- and anti-inflammatory cytokines affect endothelial functions [45,46]. The ability of immune dysregulation in cancer to produce a favorable milieu that allows for immunosurveillance escape and tumor growth has been identified as a hot topic of research. MPNs are an excellent example of inflammatory disease and a helpful model for analyzing the links between clonal proliferation, immunological tolerance loss, and chronic inflammation.

## 3. NLRP3 Protein

NLRP3 is an inflammasome molecule that can detect multiple hosts and external ligands [47]. It is a cytosolic receptor, now known as the most researched member of the inflammasome family. It is expressed in HSPCs and peripheral blood cells [48]. The *NLRP3* gene, found on chromosome 1, codes for the NLRP3 inflammasome, also known as cryopyrin, and is expressed in several cells involved in the innate immune response, including monocytes, neutrophils, lymphocytes, epithelial, and endothelial cells. The NLRP3 protein is composed of a leucine-rich repeat (LRR) domain at the C-terminus, a nucleotide-binding oligomerization (NOD or NACHT) domain in the middle, and a pyrin (PYD) domain at the N-terminus [49]. The protein structure is described in detail in Figure 1. According to a recent publication, the PYD domain is a desirable target for developing NLRP3 inhibitors due to its significance in activating the NLRP3 inflammasome [50]. After identifying the pathogens and other damage-related signals, the NLRP3 protein interacts with the ASC via its pyrin domain which binds pro-caspase-1, and converts pro-IL-1β and pro-IL-18 into their active forms. These cytokines have a pleiotropic effect on hematopoiesis, aging, and metabolic complications [51,52]. The proteolytic cleavage, maturation, and secretion of IL-1β and IL-18, as well as the cleavage of gasdermin-D (GSDMD), a specific substrate of the inflammatory caspases, are all encouraged by the inflammasome activation and assembly. This cleavage produces an N-terminal fragment that triggers pyroptosis and the generation of the cell membrane holes, causing severe membrane damage and the release of cytokines and proteins from the cytoplasm. Pyroptosis, a pro-inflammatory form of programmed cell death, affects tumor growth and aggressiveness [53,54]. NLRP3 drives this phenomenon through cell lysis and burst cell membranes. Inflammasome-mediated pyroptosis aids in the host’s defense against bacterial infections, but an unchecked process increases the risk of multiple organ failure, disseminated intravascular coagulation, and death [55]. The dying cells secrete the DAMPs, and this positive feedback loop process causes an even worse inflammatory response [56].

The NLRP3 inflammasome can be activated through a canonical, a non-canonical, and an alternative pathway. A two-step mechanism is involved in the canonical signal, in the macrophages, and the dendritic cells [57,58]. The first one, also known as the “priming step” or “signal 1”, is produced by the endogenous cytokines, PAMPs, and inflammatory stimuli such as TLR4 agonists, which cause an NF-κB-mediated NLRP3 and pro-IL-1β and pro-IL-18 expression [54]. The NF-κB-induced transcriptional priming of inflammasome proteins sets the stage for the cation channel activation, cell volume expansion, and the inflammasome component assembly [59,60]. The post-translational modifications occur during this phase. The second step, known as the “activation step” or “signal 2”, is brought on by the PAMPs, DAMPs, or glucose and amino acid efflux, which facilitates the assembly of the NLRP3 inflammasome and caspase-1-mediated IL-16 and IL-18 secretion [54]. Many NLRP3 activators induce a K+ efflux, the common trigger of the inflammasome [61]. A non-canonical and alternative pathway can turn on NLRP3 with LPS as the main trigger. The non-canonical response depends on casp-4/5 or casp-11. It has been noted that intracellular LPS directly binds to the CARD domain of casp-11 [62] and casp-4 [63], activating both, and hence these can be triggered by exogenous substances and other parts of gram-negative bacteria [64]. Both can enhance a K+ efflux, leading to the activation of the NLRP3 inflammasome, rupturing the membrane through either GSDMD cleavage and subsequent pyroptosis or currently unknown mechanisms [63,65]. In the alternative NLRP3 pathway, LPS/TLR4 alone is sufficient to cause the NLRP3 inflammasome activation through the caspase-8 signaling cascade upon TLR4/TRIF/FADD. This drives caspase-1 to become active and IL-1β to be processed and secreted. A K+ efflux is not essential for this pathway; IL-1β is secreted gradually and pyroptosis does not occur [66].

NLRP3 can be enhanced by a variety of physically and chemically unrelated stimuli. The PAMPs are microbial, fungal, viral, and parasitic products released during infection. The DAMPs are released during non-pathogen-related “sterile inflammation”, resulting from tissue/organ damage under stress [67]. The DAMPs, such as extracellular alarmins like extracellular adenosine triphosphate (eATP), nuclear protein high mobility group protein B1 (HMGB1), uric acid crystals, extracellular DNA and RNA fragments, and S100 proteins (S1009a and S1008a), trigger this activation in a paracrine/autocrine manner [68]. The release of IL-1β and IL-18 stimulates the innate immunity cells, releasing the other DAMPs and starting the complement cascade (ComC) that maintains a sterile inflammation state in the BM microenvironment [68,69,70]. NLRP3 has been suggested as a sensor for cellular homeostasis [71], and it is regulated by the GAPDH and α-enolase, mTORC1, and HK1-dependent glycolysis [72]. The DAMPs-mediated calcium influx or potassium efflux, as well as the alterations in the uptake of glucose and amino acids, all cause the activation of the NLRP3 inflammasome [73]. Among the NLRP3 modulators, NIMA-related protein kinase 7 (NEK7), along with the inhibitor of nuclear factor kappa-B kinase (IKKβ), were recently discovered to be crucial parts of the NLRP3 inflammasome and an essential modulator of the NLRP3 activity [74,75,76].

As we have seen, the NLRP3 startup takes place under different fronts. Since it is considered the undisputed protagonist, based on numerous pathologies, one of the promising goals could be to consider the NLRP3 inflammasome complex as a potential therapeutic target, especially in the hematopoietic context.

**Figure 1 ijms-24-04860-f001:**
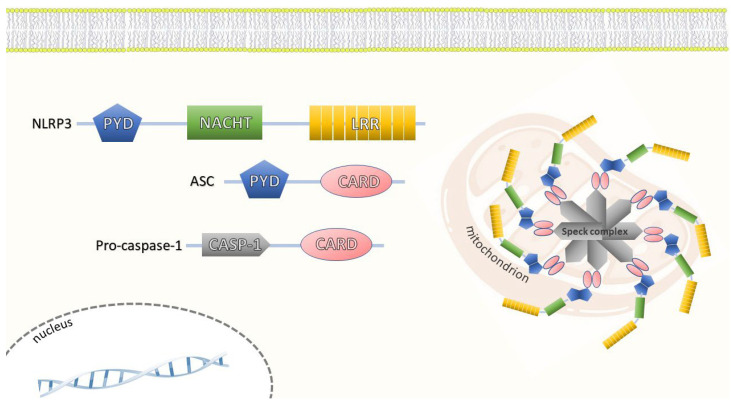
NLRP3 protein is composed of a leucine-rich repeat (LRR) domain at the C-terminus, a nucleotide-binding oligomerization (NOD or NACHT) domain in the middle and a pyrin domain (PYD) at the N-terminus [49]. The central NACHT domain provides the ATPase activity necessary for the NLRP3 activation and inflammasome formation [77]. Numerous studies revealed that the NLRP3 activity is likewise controlled by the various post-translational modifications [78] and many NLRP3-interacting proteins [79]. The NLRP3 inflammasome complex is composed of caspase-1 and an adaptor protein called apoptosis-associated speck-like protein (ASC) [47]. The ASC and pro-caspase-1 both consist of caspase activation and recruitment domains (CARD) at the C-terminus. The ASC is composed of a pyrin domain at the N-terminus while pro-caspase-1 has a casp-1 domain. By recruiting the ASC through PYD–PYD contact, the NLRP3–PYD domain is necessary to develop the active inflammasome [80]. The inactive form of this protein complex is found in the cytosol and the endoplasmic reticulum [81]. As soon as it is activated, it spreads to the mitochondria, transforming into an aggregate composed of multiple NLRP3 molecules, called “speck complexes”, each containing the NLRP3 protein, ASC, and pro-caspase 1 [65,82,83]. The ASC enables the association with the PRR component and CARD—the interaction motifs that mediate the formation of larger protein complexes—facilitating the binding between pro-caspase-1 and the PRR–ASC complex [79].

## 4. NLRP3, the Keystone in MPNs

The role of the NLRP3 protein is now evident on several fronts. While some studies proposed a preventive role for the NLRP3 inflammasome [84], most contended that it contributes to cancer pathophysiology [85]. The role of NLRP3 is evident in several neoplasms. For instance, breast cancer growth and metastasis were boosted by the NLRP3 inflammasome produced by cancer-associated fibroblasts [86]. Several hematological diseases, such as myelodysplastic syndrome (MDS), myeloproliferative neoplasms, leukemias, lymphomas and graft-versus-host diseases (GvHD), are also impacted by the NLRP3 inflammasome [48,87]. In MDS, the hematopoietic stem/progenitor cells exhibit the NLRP3 complex activation, which produces IL-1β and IL-18, causing pyroptotic cell death [88]. IL-18, triggered by the NLRP3 inflammasome, is involved in hematopoiesis and is currently thought to be primarily a proinflammatory cytokine that regulates both the innate and adaptive immunity (essential for the synthesis of IFNγ) and has a role in the etiology of autoimmune and inflammatory illnesses [27,89,90]. Through the IFNγ expression, IL-18 suppresses the development of the erythroid colonies. On the other hand, IL-1R signaling decreases erythropoietin production in the kidney [91].

Zhou et al. provided the first evidence of a higher NLRP3 inflammasome-related gene expression in MPN patients. They carried out a study on bone marrow cells for some genetic polymorphisms affecting the inflammasome genes, such as *NLRP3* (rs35829419), *NF-κB1* (rs28362491), *CARD8* (rs2043211), *IL-1β* (rs16944), and *IL-18* (rs1946518). The analysis revealed an association between MPN and a higher expression of *NLRP3*, *NF-κB1*, *CARD8*, *IL-1β*, and *IL-18*. The *NF-κB*-94 ins/del ATGG (rs28362491) polymorphism contributed to the susceptibility of MPN and to the enhancement of *NF-κB1* and the *NLRP3* expression. Given that *NF-κB* signaling hyperactivation promotes chronic inflammation in MPN, there has been substantial discussion about the therapeutic benefits of targeting NF-κB in MPN. The elevated expression of these genes was connected to the *JAK2V617F* mutation, increased white blood cell counts, and splenomegaly [92]. Unlike the other genetic variations, the JAK2 kinase activity may have a significant role in triggering the NLRP3 inflammasome [93]. The *JAK2V617F* mutation has been shown to promote the onset and development of MPN by increasing the cytokine sensitivity, constitutive activation of JAK2 kinase and the JAK/STAT signaling, and the maintenance of the cytokine-dependent survival in the cell lines [94]. However, further research is required to confirm and describe the molecular function of the NLRP3 inflammasome in *JAK2V617F*-mutant MPN [92,93,94]. The NLRP3 inflammasome’s genetic polymorphisms in CML may be used as possible outcome predictors. More specifically, the association between the polymorphisms mentioned above and their influence on the first-generation tyrosine kinase inhibitor’s (TKIs) therapeutic effects has been evaluated [95]. Many TKIs stimulate the NLRP3 inflammasome system. Among these, both imatinib and masitinib induce lysosomal expansion and cause damage that results in the cathepsin-mediated membrane instability of myeloid cells and, subsequently, cell lysis that is followed by a potassium (K+) efflux. This activates NLRP3 through the cleavage of IL-1β, caspase-1, and GSDMD, as well as the formation of the ASC specks. This effect is restricted to the primary myeloid cells (such as the peripheral blood mononuclear cells and the mouse bone marrow-derived dendritic cells) [96]. How the TKIs cause lysosomal instability is still unclear, but it is known that certain TKIs, including imatinib, have been found to accumulate in the lysosome [97,98] causing an increased osmolarity, enlargement, and possible rupture [99]. Neuwirt et al., using the membrane stabilizer polyethylene glycol (PEG), discovered that lysosomal loss is insufficient for activating NLRP3, but that the TKIs play an important role in triggering lytic cell death and the inflammasome signals [96,100]. Studies on the cells lacking caspase-1, or GSDMD, have shown that imatinib and masitinib induce an inflammasome-independent form of lytic cell death and a K+ efflux–dependent NLRP3 activation [96].

According to several studies on myeloid lineage, the mature and progenitor cells exhibit different cytokine profiles, indicating that they play separate roles in MPN pathogenesis [30]. However, Zhou et al. showed that both cancerous MPN cells and their healthy counterparts triggered inflammatory responses. It can, therefore, be presumed that the NLRP3 inflammasome plays a role in both healthy and malignant hemopoietic cells [92]. As mentioned above, the development of MPN is accelerated by excessive inflammatory signaling pathways, such as NF-κB and STAT, which lead to aberrant inflammatory cytokine production and ongoing immune cell overreaction [101]. The NLRP3 inflammasomes’ importance in MPN is poorly understood, even if it is known that inflammatory cytokines that encourage clonal expansion to extramedullary locations are directly linked to splenomegaly development [102]. This finding showed that patients with a greater symptom burden had increased NLRP3 inflammasome activity. An increased release of IL-1β and IL-1R signaling from the NLRP3 inflammasome enhanced the production of pro-myelopoietic cytokines in BM accessory cells, explaining the rise in myelopoiesis, at least in part [103].

Another key player in the MPNs inflammatory scenario is hepcidin, a negative regulator of iron homeostasis, whose synthesis is stimulated by the ROS, which also promotes the release of IL-1β in human monocytes by activating NLRP3. A functional iron deficiency is common in MF where immunological dysregulation and abnormal inflammatory cytokine production lead to an increase in hepcidin. Hepcidin reduces the bioavailable iron by inhibiting intestinal absorption, downregulating the iron exporter channel ferroportin, and increasing iron deposits in the monocyte–macrophage system. As a result of this mechanism, hyperinflammation will unavoidably become worse. Birgegard et al. discovered that the inflammatory state of myelofibrosis affects the iron turnover and plays a role in the development of anemia [104].

Researchers have also found a connection between the MPN-related NLRP3 inflammatory process and the micro-RNA miR-146a. miR-146a-5p has been identified as a negative regulator of the innate immunological and inflammatory responses mediated by Toll-like receptor 4. In mice, the miR-146a wild type significantly suppressed autoimmune disease, myeloproliferation, and cancer [105], and, in the context of the TLR4 pathway, it negatively regulated the innate immunological and inflammatory responses [106]. Compared to the controls, the rs2431697 TT genotype was frequently found in MPN patients with MF subtypes. The polymorphism is considered a marker for MF early progression. The study revealed that the TT genotypes were linked to an elevated expression of inflammation-related genes, specifically *NLRP3*, *NF-κB1*, and *IL-1β*. The elevated expression of these genes in the BM cells from the MPN patients was related to the *JAK2V617F* mutation, white blood cell counts, and splenomegaly [105,106,107].

The patients with *KRAS* mutations presented a higher caspase-1 activation and IL-1β production than those with the *KRAS* wild type in CMML, JMML, and AML. The microarray-based studies revealed that the *NLRP3* expression was elevated in murine hematopoietic bone marrow cells carrying the active inducible *KrasG12D* allele. Compared to the wild-type the *KrasG12D* BM-derived dendritic cells produced more IL-1β and triggered caspase-1, supporting NLRP3’s functional relevance in the myeloid compartment. *KrasG12D* mice lacking NLRP3 in the hematopoietic system did not exhibit cytopenia or myeloproliferation, unlike those that have expressed it. This demonstrates that oncogenic *KrasG12D* initiates the RAC1/ROS/NLRP3/IL-1β axis, a potential target for the treatment strategies that regulate myeloproliferation. Through the expression analysis, Shaima et al. demonstrated that the KRAS/RAC1 pathway triggered NLRP3 and produced the ROS. This suggests that oncogenic *KRAS* affects the NLRP3/IL-1β axis via its oncogenic driver function but also increases its activation [108]. The inflammasome scenery in the MPN cells is illustrated in Figure 2. Studies on the relationship between NLRP3 and MPNs are still lacking but given the significant evidence of this protein in hematological diseases and cancer in general, there are excellent reasons to investigate the role of NLRP3 in the MPNs on several fronts. It might be interesting to get more substantial data on the NLRP3 multi-protein complex from a biochemical and structural point of view and its interactions with the inflammasome molecules in the MPNs context. Given the biochemical heterogeneity of MPNs, it would also be interesting to study the real effect of the therapeutic combination of the JAK2 and NLRP3 inhibitors on a large cohort of patients who are refractory to treatment.

## 5. Novel Therapeutic Approaches

There are limits to the JAK inhibitors’ activity despite the radical changes they have brought to the MPN landscape and their crucial part in the treatment of MF [109]. Several intriguing new drugs such as BET inhibitors (pelabresib), BcL-xl inhibitors (navitoclax), and PI3K inhibitors (parsaclisib), with different mechanisms of action beyond the JAK-STAT pathway, are in advanced clinical development. These can be used alone or with ruxolitinib, a targeted JAK2 inhibitor [110,111]. In the preclinical investigations, novel immunotherapies have been investigated, including neoepitope-directed vaccinations and monoclonal antibodies against mutant-driven MPNs [112,113]. The JAK2-NLRP3 axis has been studied in vitro in autoimmune inflammatory diseases in which JAK modulates the myelination/demyelination balance in the neurons, at least through the NLRP3-mediated pathways. The ruxolitinib-inhibited *NLRP3* expression, phosphorylation of JAK2, and IL-1β are released, induced by the thymic stromal lymphopoietin receptor [114]. Zhu et al. demonstrated that the JAK2 inhibition through ruxolitinib administration reduces the NLRP3 inflammasome activation through the JAK2/STAT3 pathway, to improve the ischemic stroke damage and neuroinflammation. Ruxolitinib suppresses the production of the NLRP3 inflammasome components and reduces several proinflammatory cytokines [115]. The novel therapeutic agents target various biomolecules from the inflammasome pathway in MDS and AML [116]. Inflammasome targeting therapies can be explored as combinatorial strategies with JAK2 inhibitors as possible synergistic mechanisms, but we have no data about this in the MPN context.

## 6. Conclusions

The world of the inflammasome is fascinating because it is the basis of the most common neoplasms. Studies of healthy and unhealthy hematopoiesis have transformed the NLRP3 inflammasome into an intriguing topic [48]. The relationship between NLRP3 and MPNs has now become evident. Since inflammasomes play a role in myeloid malignancies, they are potentially appealing therapeutic targets. Several NLRP3 inhibitors have been created, and some are currently undergoing clinical trials to treat cancer and inflammatory diseases [84]. The study of genetic variations, such as the copy number variants (CNVs), indels (deletions or insertions), structural variants, and single nucleotide polymorphisms (SNPs), further increased since the development of high-throughput techniques and has greatly aided the diagnosis and treatment of diseases [92].

Although the biochemical heterogeneity of MPNs has not yet been fully understood, current knowledge about the function of inflammasomes is encouraging the creation of novel therapeutic approaches. Combination therapies that eliminate uncontrolled proliferation, systemic inflammation, and loss of immunoregulation will soon dominate. To ensure a more significant and accurate assessment of these disorders and their care, hematologists and oncologists should develop interdisciplinary expertise. For targeted therapy, translational research should also examine the relationships between the clinical manifestations, risk scores, molecular profiles (and its evolution), and the participation of the inflammasomes [117]. Although inflammasome-targeting immunotherapies in hematology have not yet entered clinical use, the wide range of interactions opens up new possibilities for disease management [87].

It is known that the administration of the NLRP3 inhibitors reduced the severity of the disease in AML [118], DLBCL [119], GvHD [120], multiple myeloma [121], and sickle cell anemia [122] in both vitro and in vivo studies. Starting from the protein, the PYD domain is an attractive target for developing the NLRP3 inhibitors due to its importance in the NLRP3 activation [50]. MCC950, a selective NLRP3 inhibitor, could be a promising drug candidate to stop the advancement of myeloid malignancies caused by NLRP3-mediated illness [123]. Myeloproliferation was decreased through therapy with either the IL-1β receptor blockade or MCC950. Other potent novel medications are being researched, and the first human clinical trials will soon begin [48]. Today, MCC950 has demonstrated its universality by preventing inflammasome-induced platelet aggregation in sickle cell anaemia [124]. In the hematological context, ibrutinib is the BTK inhibitor that binds specifically to the ASC and NLRP3, preventing the inflammasome activation [125]. Ibrutinib-like substances can also decrease the IL-1β synthesis by suppressing caspase-1 [126]. This substance is being tested in high-risk MDS phase I clinical studies [127]. Through regulation of the miRNA rs2431697 genotype and *NLRP3*, *NF-κB1*, and *IL-1β* genes, new therapeutic strategies could be considered to prevent myelofibrosis progression in MPN patients [107]. For *KRAS*-driven hematological malignancies, a therapeutic approach might include the use of NLRP3 and IL-1R suppressors [108].

Inflammasomes can contribute to the pathophysiology, development, and progression of cancer. They may build and maintain the tumor microenvironment in some kinds of neoplasms. Given the good evidence of NLRP3 inhibitors in blood malignancies, although there is still much to learn about the variability of MPNs, studying the NLRP3 multi-protein complex from a biochemical and structural point of view and its interactions with the inflammasome molecules could pave the way for several novel therapeutic options in the world of MPNs.

## Figures and Tables

**Figure 2 ijms-24-04860-f002:**
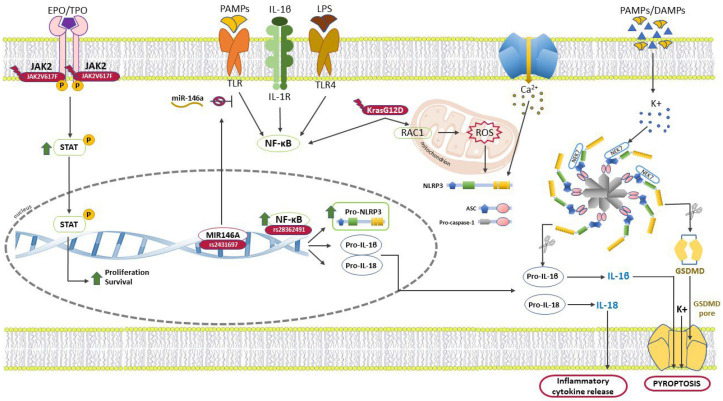
MPN is accelerated by the overactive inflammatory STAT and NF-κB1 signaling pathways. The MPN patients’ BM cells with the *JAK2V617F* mutation showed a high expression of the *NLRP3*, *NF-κB1*, *IL-1β*, *IL-18*, and *CARD8* genes. The NLRP3 inflammasome priming step was activated by the PAMPs, LPS and cytokines through the NF-κB1 pathway. In the nucleus, this molecule promoted the transcription of *NLRP3*, *IL-1β* and *IL-18*. The *NF-κB-94* ins/del ATGG (rs28362491) polymorphism was related to the enhancement of *NF-κB1* and the *NLRP3* expression. The inflammasome assembly (activation step) was brought on by a Ca2+ efflux and the PAMPs and DAMPs through a K+ efflux increase. NLRP3 combined with the ASC and caspase-1 proteins to form the spike complex. The spike cleaved pro-IL-1β, pro-IL-18 and GSDMD into their active form. GSDMD pore forming, along with the K+ efflux and IL-1β protein, gave rise to pyroptosis. The *KrasG12D* mutation enhanced the NLRP3/IL-1β axis and also increased the NLRP3 activation, triggering the RAC1/ROS signaling. Normally, the miR-146a protein negatively regulates the innate immunological and inflammatory responses. Otherwise, the expression of the *miR-146a* rs2431697 TT genotype was frequently found in MPNs patients and was linked to an elevated expression of *NLRP3*, *NF-κB1*, and *IL-1β* inflammatory genes. NLRP3: nucleotide-binding domain-like receptor protein 3; EPO: erythropoietin; TPO: thrombopoietin; PAMPs: pathogen-associated molecular patterns; DAMPs: danger-associated molecular patterns; LPS: lipopolysaccharide; TLR: toll-like receptor; GSDMD: gasdermin-D.

## Data Availability

Not applicable.

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
