# Peer review of "The Role of NLRP3, a Star of Excellence in Myeloproliferative Neoplasms"

_ijms, 2023, doi:10.3390/ijms24054860_

Round 1
Reviewer 1 Report
Dr. Parciante et al. have put together a well-written, thoughtfully organized, and a comprehensive review on an exciting topic of the role of NLRP3 in heme malignancies with a focus on MPNs.
I just have a few suggestions to consider. I don’t have any major revisions, just minor edits:
1. Should consider including the protective role of NLRP3 inflammasome in responding to infections? When considering therapeutic interventions, this needs to be addressed as a potential toxicity.
2. On page 4, line 199 – I think the author mean “context” instead of “contest”
3. It may help to put in context if a paragraph on current therapies for MPN specifically JAK2 inhibitors since it is upstream to NLRP3 assembly and how NLRP3 blockade may be different – more targeted?
4. Any suggestion from the literature whether this pathway participates more or less in myelofibrosis?
5. Authors mention on page 7, line 315 “excellent reasons to investigate the role of NLRP3 in the MPNs on several fronts…” Provide some examples or suggestions as to what is the future direction
6. Page 3 line 130 – redundant “most closely studied ever” and “most researched member of the inflammasome family” – remove one
Author Response
We would like to thank the Reviewer for the helpful suggestions. The manuscript has been revised according to the following comments.
Point 1: “Should consider including the protective role of NLRP3 inflammasome in responding to infections? When considering therapeutic interventions, this needs to be addressed as a potential toxicity”
Response 1: As suggested, we added a sentence (lines 62-66) in which we considered the protective role of NLRP3 inflammasome.
Point 2: “On page 4, line 199 – I think the author mean “context” instead of “contest””
Response 2: As suggested, the text has been modified (line 210).
Point 3: “It may help to put in context if a paragraph on current therapies for MPN specifically JAK2 inhibitors since it is upstream to NLRP3 assembly and how NLRP3 blockade may be different – more targeted?”
Response 3: A new paragraph about JAK2 inhibitors in the inflammasome context has been added, as suggested (line 374)
Point 4: “Any suggestion from the literature whether this pathway participates more or less in myelofibrosis?”
Response 4: The information reported in the literature about the NLP3 involvement in MF pathogenesis are cited in lines 307-326.
Point 5: “Authors mention on page 7, line 315 “excellent reasons to investigate the role of NLRP3 in the MPNs on several fronts…” Provide some examples or suggestions as to what is the future direction”
Response 5: We added some suggestions in the paragraph (lines 347-352)
Point 6: “Page 3 line 130 – redundant “most closely studied ever” and “most researched member of the inflammasome family” – remove one”
Response 6: As suggested, the text has been modified (line 139)
Reviewer 2 Report
This review manuscript has shed lights on the intriguing NLRP3 inflammasome complex that has become a rising star in studies focused on pathological hematopoiesis, such as myeloproliferative neoplasms (MPNs). Deep understanding of the role of NLRP3 is fundamental to advance therapeutic targeting strategies in MPNs patients. This review manuscript provided with a through introduction of the target (NLRP3) and the disease (MPNs), as well as current knowledge about the role of NLRP3 in MPNs. This review manuscript is well written, and the reference is appropriate and adequate. Additionally, the discussion feature prominently.
Please address the following issues before acceptance.
(Line 29) The statement of MPNs characterized by a mutation of the HSPCs needs to be modified. Genetically, MPNs are very heterogeneous and characterized by uncontrolled proliferation of abnormal HSPCs, with a wide variety of genetic abnormalities. As described in the following paragraph, the JAK2/MPL/CALR "driver" alterations, and additional mutations co-exist in most MPNs.
(Line 60) Some studies, including the Ref 19, suggest that the involvement of inflammasome activation may induce clonal selection of mutated HSPCs and thus drive the progression of MPNs. Here, the authors stated that the genetic abnormalities in MPNs seems contributing to inflammasome activation, which is an opposite opinion. Please clarify it as needed.
(Line 80) Please correct the statement based on Ref 27. Ablation of IL-1β in MNP mice did NOT support the clonal proliferation of JAK2 mutated HSPCs. In fact, mice transplanted with VF;IL1β-/- bone marrow cells showed less engraftment, which supported that IL1β favors clonal expansion during MPN disease initiation.
(Line 240) Please add the ref and clarify what is the link exactly demonstrated in the paper. Is the link between NK-κB and NLRP3 expression or NK-κB/NLRP3 and MPN susceptibility?
(Line 233) To address the role of NLRP3 in MPNs, authors firstly talked about the first evidence of NLRP3 inflammasome-related gene expression in MPN patients. However, the following paragraph (Line 242-267) was not clearly presented. There seems lots of different things that the authors were trying to talk about, like 1) NF-κB is a potential target in MPN (or NF-κB signaling activated NLRP3 inflammasome?). 2) JAK2 mutation may trigger NLRP3 inflammasome. 3) NLRP3 inflammasome is potential predictor/biomarker of clinical outcome of TKIs in CML patients. I highly recommend reorganizing this paragraph and rephasing some statements to avoid redundancy.
(Line 283-290) The authors were discussing about the role of NLRP3 in MPNs inflammatory scenario. Regarding hepcidin, it is obscure what authors want to conclude. (Line 284) Hepcidin promotes the NLRP3 dependent IL-1β release. (Line 286) The abnormal inflammatory cytokine production (e.g., IL-1β) lead to an increase in hepcidin.
(Line 291-299) Please clarify the connection/association between NLRP3 inflammatory process and miR-146a? MiR-146a rs2431697 polymorphism is linked to the elevated expression of NLRP3 etc (Line 294). It is confusing that the increase in NLRP3 etc. expression contributes to the decreasing of miR-146a expression (Line 296).
Author Response
We would like to thank the Reviewer for the helpful suggestions. The manuscript has been revised according to the following comments.
Point 1: “(Line 29) The statement of MPNs characterized by a mutation of the HSPCs needs to be modified. Genetically, MPNs are very heterogeneous and characterized by uncontrolled proliferation of abnormal HSPCs, with a wide variety of genetic abnormalities. As described in the following paragraph, the JAK2/MPL/CALR "driver" alterations, and additional mutations co-exist in most MPNs.”
Response 1: As suggested, the text has been modified (line 30).
Point 2: “(Line 60) Some studies, including the Ref 19, suggest that the involvement of inflammasome activation may induce clonal selection of mutated HSPCs and thus drive the progression of MPNs. Here, the authors stated that the genetic abnormalities in MPNs seems contributing to inflammasome activation, which is an opposite opinion. Please clarify it as needed.”
Response 2: We clarified the concept of the dual role of NLRP3 in lines 62-66.
Point 3: “(Line 80) Please correct the statement based on Ref 27. Ablation of IL-1β in MNP mice did NOT support the clonal proliferation of JAK2 mutated HSPCs. In fact, mice transplanted with VF;IL1β-/- bone marrow cells showed less engraftment, which supported that IL1β favors clonal expansion during MPN disease initiation.”
Response 3: The text has been modified to express the concept better (line 86).
Point 4: “(Line 240) Please add the ref and clarify what is the link exactly demonstrated in the paper. Is the link between NK-κB and NLRP3 expression or NK-κB/NLRP3 and MPN susceptibility?”
Response 4: The paragraph from lines 250 to 261 has been rearranged to clarify better.
Point 5: “(Line 233) To address the role of NLRP3 in MPNs, authors firstly talked about the first evidence of NLRP3 inflammasome-related gene expression in MPN patients. However, the following paragraph (Line 242-267) was not clearly presented. There seems lots of different things that the authors were trying to talk about, like 1) NF-κB is a potential target in MPN (or NF-κB signaling activated NLRP3 inflammasome?). 2) JAK2 mutation may trigger NLRP3 inflammasome. 3) NLRP3 inflammasome is potential predictor/biomarker of clinical outcome of TKIs in CML patients. I highly recommend reorganizing this paragraph and rephasing some statements to avoid redundancy.”
Response 5: As mentioned above, the paragraph from lines 250 to 272 has been rearranged.
Point 6: “(Line 283-290) The authors were discussing about the role of NLRP3 in MPNs inflammatory scenario. Regarding hepcidin, it is obscure what authors want to conclude. (Line 284) Hepcidin promotes the NLRP3 dependent IL-1β release. (Line 286) The abnormal inflammatory cytokine production (e.g., IL-1β) lead to an increase in hepcidin.”
Response 6: The text has been modified to explain better what the authors want to conclude (lines 307-314)
Point 7: “(Line 291-299) Please clarify the connection/association between NLRP3 inflammatory process and miR-146a? MiR-146a rs2431697 polymorphism is linked to the elevated expression of NLRP3 etc (Line 294). It is confusing that the increase in NLRP3 etc. expression contributes to the decreasing of miR-146a expression (Line 296).”
Response 7: To better clarify, the text has been modified (lines 316-326).